# Hemochromatosis: Ferroptosis, ROS, Gut Microbiome, and Clinical Challenges with Alcohol as Confounding Variable

**DOI:** 10.3390/ijms25052668

**Published:** 2024-02-25

**Authors:** Rolf Teschke

**Affiliations:** 1Department of Internal Medicine II, Division of Gastroenterology and Hepatology, Klinikum Hanau, D-63450 Hanau, Germany; rolf.teschke@gmx.de; Tel.: +49-6181-21859; Fax: +49-6181-2964211; 2Academic Teaching Hospital of the Medical Faculty, Goethe University Frankfurt/Main, D-60590 Frankfurt am Main, Germany

**Keywords:** hemochromatosis, iron overload, serum ferritin, fasting transferrin saturation index, genetic testing, family screening, *HFE* genes, phlebotomy, alcohol abuse, Haber–Weiss reaction, Fenton reaction, ROS, ferroptosis, gut microbiome

## Abstract

Hemochromatosis represents clinically one of the most important genetic storage diseases of the liver caused by iron overload, which is to be differentiated from hepatic iron overload due to excessive iron release from erythrocytes in patients with genetic hemolytic disorders. This disorder is under recent mechanistic discussion regarding ferroptosis, reactive oxygen species (ROS), the gut microbiome, and alcohol abuse as a risk factor, which are all topics of this review article. Triggered by released intracellular free iron from ferritin via the autophagic process of ferritinophagy, ferroptosis is involved in hemochromatosis as a specific form of iron-dependent regulated cell death. This develops in the course of mitochondrial injury associated with additional iron accumulation, followed by excessive production of ROS and lipid peroxidation. A low fecal iron content during therapeutic iron depletion reduces colonic inflammation and oxidative stress. In clinical terms, iron is an essential trace element required for human health. Humans cannot synthesize iron and must take it up from iron-containing foods and beverages. Under physiological conditions, healthy individuals allow for iron homeostasis by restricting the extent of intestinal iron depending on realistic demand, avoiding uptake of iron in excess. For this condition, the human body has no chance to adequately compensate through removal. In patients with hemochromatosis, the molecular finetuning of intestinal iron uptake is set off due to mutations in the high-FE^2+^ (*HFE*) genes that lead to a lack of hepcidin or resistance on the part of ferroportin to hepcidin binding. This is the major mechanism for the increased iron stores in the body. Hepcidin is a liver-derived peptide, which impairs the release of iron from enterocytes and macrophages by interacting with ferroportin. As a result, iron accumulates in various organs including the liver, which is severely injured and causes the clinically important hemochromatosis. This diagnosis is difficult to establish due to uncharacteristic features. Among these are asthenia, joint pain, arthritis, chondrocalcinosis, diabetes mellitus, hypopituitarism, hypogonadotropic hypogonadism, and cardiopathy. Diagnosis is initially suspected by increased serum levels of ferritin, a non-specific parameter also elevated in inflammatory diseases that must be excluded to be on the safer diagnostic side. Diagnosis is facilitated if ferritin is combined with elevated fasting transferrin saturation, genetic testing, and family screening. Various diagnostic attempts were published as algorithms. However, none of these were based on evidence or quantitative results derived from scored key features as opposed to other known complex diseases. Among these are autoimmune hepatitis (AIH) or drug-induced liver injury (DILI). For both diseases, the scored diagnostic algorithms are used in line with artificial intelligence (AI) principles to ascertain the diagnosis. The first-line therapy of hemochromatosis involves regular and life-long phlebotomy to remove iron from the blood, which improves the prognosis and may prevent the development of end-stage liver disease such as cirrhosis and hepatocellular carcinoma. Liver transplantation is rarely performed, confined to acute liver failure. In conclusion, ferroptosis, ROS, the gut microbiome, and concomitant alcohol abuse play a major contributing role in the development and clinical course of genetic hemochromatosis, which requires early diagnosis and therapy initiation through phlebotomy as a first-line treatment.

## 1. Introduction

Heavy metals are all of extraterrestrial origin, formed in the universe from helium and hydrogen via nuclear fusion in stars before arriving at earth and in humans [1,2,3,4]. Some of them are risky for exposed humans if accumulated by human activities, as these metals are present almost anywhere as pollutants in the environment [5,6]. As an example, cadmium can easily and constantly enter the human body, which cannot provide cadmium homeostasis due to a lack of physiological processes to remove it [6]. The conditions are different for iron and copper because they are not useless elements but become essential trace metals for humans, ensuring health and life, although copper in excess can lead to the copper-storing liver disorder known as Wilson’s disease [7,8], while an overload of iron may cause the hepatic iron-storing disease hemochromatosis [9,10,11,12].

In this article, the focus is on the mechanistic sequalae that lead to hereditary hemochromatosis and on its clinical features of diagnosis and therapy. Discussed topics include ferroptosis, reactive oxygen species (ROS), the gut microbiome, and the mechanistic role of alcohol abuse as a confounding variable in hemochromatosis. In addition, other iron-related liver diseases are briefly discussed, which are caused by an intentional uptake of large exogenous iron amounts or by an unusual internal release of iron from erythrocytes in the course of genetic hemolytic diseases that cause a secondary hepatic iron overload form different from hereditary hemochromatosis.

## 2. Role of Iron in Human Health

Iron absorbed by enterocytes from consumed food and beverages in normal amounts is involved in abundant physiological processes that help sustain health in humans (Table 1) [13,14,15].

Humans and other organisms like animals or plants cannot synthesize iron and consequently depend on its uptake. For instance, plants commonly absorb iron contained in soil [16], and cattle receive iron by grazing on fields. Finally, humans receive their iron via the food of plants like vegetables and by consuming meat [15].

## 3. Iron Deficiency and Iron Overload

In humans, reduced total body iron stores due to low iron uptake with food and beverages primarily leads to an impaired generation of red blood cells, causing anemia with fatigue [13,14,15]. On the other hand, unphysiological high total body iron stores are achieved by exogenous iron uptake in large amounts mostly by intention via the gastrointestinal tract [17,18]. Total body iron stores are found to be elevated in a variety of genetic hemolytic disorders [19]. These conditions are found in patients with thalassemia, sickle cell anemia, and hereditary spherocytosis, all often associated with multiple transfusions of red blood cells to treat anemia or if iron is liberated from hemoglobin during hemolysis in cases of genetic causes [19]. Increased total body iron stores are also hallmarks of hemochromatosis [9,10,11,12], with iron being found preferentially in the liver but also less in other organs [20]. The causes of hepatic iron overload are variable (Figure 1).

## 4. Factors Ensuring Sustained Iron Homeostasis

Humans must take up iron from the food chain and from beverages that contain this trace element, but they benefit from their own internal mechanisms, ensuring that iron homeostasis is achieved through various essential steps. Under physiological conditions, healthy individuals allow for iron homeostasis by restricting the extent of intestinal iron absorption depending on realistic demand, avoiding the uptake of iron in excess [13,14,15,21]. As an unregulated process, iron excretion occurs through loss in sweat, menstruation, and shedding of hair and skin cells, which is associated with rapid turnover and the excretion of enterocytes [13,14,22].

Dietary iron arrives at the intestinal tract as heme iron originating from animal food sources like meat, seafood, and poultry or as non-heme iron mainly derived from plants [22]. However, to be absorbable, insoluble ferric (Fe^3+^) iron must be reduced to absorbable ferrous (Fe^2+^) iron within the intestinal lumen. This is carried out by the ferric reductase known as duodenal cytochrome B (Dcytb), an apical membrane-bound enzyme of enterocytes functioning to reduce insoluble ferric iron [22]. The subsequent intestinal uptake of absorbable iron occurs preferentially in the duodenum in humans [23,24], confirming previous data in animals that were used for experimental iron-related studies [25,26]. At the cellular level, divalent metal cation transporter 1 (DMT1), a protein on the apical membrane of enterocytes, transports iron across the apical membrane into the cell [22]. Activity levels of DMT1 and Dcytb are upregulated in the hypoxic environment of the intestinal mucosa by hypoxia-inducible factor-2 (HIF-2α) [22,27,28].

At the molecular level and once inside the enterocytes, iron can be stored as ferritin if not needed [22]. Alternatively, if iron is later required in the human body, it can actively be transported out of the enterocyte into the circulation across the basolateral membrane of the enterocyte through the transmembrane protein iron exporter ferroportin [13,22], which is regulated by hepcidin levels that are increased by high levels of iron, inflammatory cytokines, and oxygen [22]. Hepcidin is a liver-derived peptide that impairs the release of iron from enterocytes and macrophages by interfering with ferroportin [29].

The human hemochromatosis high-FE^2+^ (*HFE*) gene, linked to the major histocompatibility complex (MHC) on chromosome 6p, encodes the MHC class-I-like protein *HFE* that binds beta-2 microglobulin [30]. *HFE* influences iron absorption by modulating the expression of hepcidin in the liver, the main key player and controller of iron metabolism [13,21,30].

## 5. Hemochromatosis Caused by Iron as a Toxic Element

### 5.1. Pathophysiology

*HFE*-associated hemochromatosis is genetically characterized by mutations in hemochromatosis (*HFE*) genes [13]. As a result, hepcidin levels are reduced or not measurable [24,29,30,31,32]. In addition, resistance on the part of ferroportin to hepcidin binding may be observed. These alterations are viewed as major mechanisms for the excessive intestinal iron uptake and the subsequent increased iron stores in the body [13,24,29,30,31,32]. Hepcidin is known as a key iron modulator under the control of ferroportin, the cellular protein in enterocytes exporting iron out of the cell [32]. Consequently, low plasma levels of hepcidin cause elevated ferroportin levels in enterocytes and consequently lead to a high intestinal iron uptake and iron overload [32]. In patients with hemochromatosis, the molecular finetuning of physiological intestinal iron uptake is set off primarily due to genetic mutations in the *HFE* gene that cause an increased absorption of iron despite a normal dietary iron intake.

### 5.2. Natural Course

Hemochromatosis is a genetically heterogeneous disorder [33,34]. It causes iron overload by the excess intestinal absorption of dietary iron due to a decreased expression of intestinal hepcidin [35]. Excess amounts of iron are primarily found in the liver with its hepatocytes, which comprise around 80% of the liver mass and are able to synthesize a high amount of the iron storage protein ferritin [14]. Iron is also stored in other parenchymal cells like pancreatic cells and cardiomyocytes [36] as well as joints, thyroid, skin, gonads, and pituitary [37].

### 5.3. Prevalence

Hemochromatosis is known as a Celtic disease and is tightly linked to mutations within the *HFE* gene C282Y [38,39,40,41]. This mutation is preferentially detected in populations of northern European ancestry with a Celtic origin [37,38,39,40,41,42]. The highest European frequencies were observed in the Celtic populations in Ireland, the United Kingdom, and France but were also detected in Scandinavia of Viking origin [42] and in Busselton, a town in western Australia with residents predominantly of Anglo-Celtic descendance as immigrants from the United Kingdom [43].

Hemochromatosis occurs in homozygotes with a mutation of the *HFE* gene at a prevalence of 1:300 to 1:500 individuals [34]. C282Y and H63D are the most common mutations of the *HFE* gene, present on the short arm of chromosome 6 (6p21.3) [34,44]. Different types of hemochromatosis are currently under discussion [33,34], now focusing on the most recent proposal (Table 2) [34,45,46,47].

Hemochromatosis with its retained iron primarily deposited in the hepatocytes is to be differentiated from hepatic iron overload syn hemosiderosis with its primarily deposited iron as hemosiderin in the reticuloendothelial cells of the liver [19,34]. This form of hepatic iron overload can occur in a variety of hemolytic conditions. Among these are anemia with ineffective erythropoiesis like thalassemia, sickle cell anemia, and hereditary spherocytosis, the overall conditions of which are often associated with multiple transfusions of red blood cells to treat anemia [19]. If iron overload is severe, chelation as a first-line therapy is indicated to avoid disease progress. Of course, venesection is not possible due to already existing anemia.

### 5.4. Clinical Features

The initial symptoms of hemochromatosis are uncharacteristic. These include asthenia, joint pain, arthritis, chondrocalcinosis, diabetes mellitus, hypopituitarism, hypogonadotropic hypogonadism, cardiopathy, heart failure, and cardiac arrhythmias [33]. Accordingly, patients are often seen by non-hepatologists. More specifically, general practitioners, orthopedists, diabetologists, cardiologists, endocrinologists, and dermatologists for skin pigmentation are rarely seen and only after a long disease course. In addition, koilonychia, a nail disease that causes thin, flat, or concave nails, affecting the thumb and index finger has been observed in 50% of patients, while 25% of patients tend to have it in all nails [34]. The male gender is preferred, with an odds ratio of 4.3 (95% CI 2.97–6.18) for the liver disease [48], as pre-menopausal females lose iron through blood physiologically via menstruation. This results in delayed disease detection in females, reported often in their sixth decade as compared with males in their fifth decade [34].

In hemochromatosis studied by histopathology, the first signs of the liver injury is commonly steatosis, or it is associated with mild inflammatory features in the sense of steatohepatitis [49,50]. Unless the disease is treated early in this stage, the injury progresses to chronic liver disease, as evidenced by fibrosis, cirrhosis, and eventually hepatocellular carcinoma (HCC) [50]. The initial stages of the disease are clinically not easily detectable by conventional laboratory parameters or apparent symptoms. Similarly, even when cirrhosis develops, it is often a silent cirrhosis lacking symptoms before clinical signs of decompensated cirrhosis emerge like jaundice, ascites, bleeding esophageal varices, and hepatic encephalopathy [49]. The end stages of the natural course are acute liver failure, hepatocellular carcinoma, or death [48,49].

Using electron microscopy for assessing the liver of patients with hemochromatosis, injurious changes in mitochondria and the endoplasmic reticulum were described in necrotic hepatocytes with the highest liver iron content, associated with large iron-laden lysosomes in the hepatocytes that were also encountered in the Kupffer cells [51]. In the pre-cirrhotic stage of hemochromatosis, the first evident ultrastructural changes are found in the lysosomal compartment [51]. These changes also correlate well with the iron overload in the advanced stages of the disease and are reversable after therapeutic iron removal from the blood.

### 5.5. Diagnosis

With increasing iron accumulation in the liver or difficult interpretable laboratory data, iron quantification can easily be achieved by using magnetic resonance imaging (MRI) [52]. In the early stage of hemochromatosis, serum LTs like ALT and AST are normal or marginally increased up to 2 times the upper limit of normal (ULN) [34,53]. Later disease stages such as cirrhosis alone or combined with hepatocellular carcinoma (HCC) are clinically easier to recognize and may provide higher values of ALT, AST, and ALP, especially in the stage of decompensated cirrhosis [53]. Considering that mild abnormalities in the biochemical liver profile are common in hemochromatosis, patients with an unexplained abnormality in the liver profile should be screened for hemochromatosis using serum parameters and genetic testing [53,54].

Serum parameters such as ferritin and fasting transferrin saturation evaluating the normal and pathological total body iron status or single segments thereof are highly variable. The variability can be traced back to age, gender, and laboratory ranges of normal that may differ from among various laboratories depending on the analytical method used and the definition of a cutoff [53,54]. The determination of serum iron lacks clinical relevance due to its variability and missing specificity. Instead, among the most used parameters in evaluating iron status is serum ferritin in combination with fasting transferrin saturation, despite the remaining issues of methodology [53,54,55]. Cytosolic ferritin is responsible for intracellular iron storage and for overall iron homeostasis by entering the blood, where it supplies various other organs with iron as serum ferritin that is delivered if demand exists [55,56]. As a result, serum ferritin is basically a marker of the total iron body store, provided that other confounding variables that also increase serum ferritin levels are carefully excluded. This is mandatory, as ferritin is a typical acute-phase protein that is increased in inflammatory conditions, thus confounding the interpretation of obtained high values [53,54,55,56,57,58].

Empirically, the combination of the three parameters of high values of serum ferritin, elevated fasting transferrin saturation, and genetic screening may help to suspect hemochromatosis [33]. A high serum ferritin level alone is, however, not sufficient to establish the diagnosis of hemochromatosis. High levels are often confounded by a variety of other diagnoses. The most common confounders are liver diseases like non-alcoholic steatohepatitis (NASH) or viral hepatitis, alcohol excess, infections, malignancy, renal failure, and metabolic syndrome, while less common causes are thyrotoxicosis and acute myocardial infarction [59,60]. As a result, a cautionary statement is warranted because only up to 10% of patients with increased serum ferritin levels may have some kind of iron overload, and this is not ascertained in 90% [58]. Iron overload in patients with alcoholic liver disease associated with increased serum ferritin levels is a common observation in clinical practice [58,60,61,62]. To assess the causative event leading to increased serum ferritin levels in this patient cohort, attempts are warranted to analyze this parameter under conditions of recommended alcohol abstinence. Normalization will exclude hemochromatosis, while persisting high ferritin levels will provide evidence of hemochromatosis or continued alcohol consumption. This must be verified by assessing serum gamma-glutamyltransferase activities as diagnostic markers.

Serum transferrin binds to iron after transportation through the basolateral membrane of enterocytes [58] and plays a diagnostic role as a transferrin saturation index [33,34,63,64]. Values of more than 40% in women and more than 50% in men should lead to further testing, and similarly if serum levels of ferritin are above 200 µg/L in women and above 300 µg/L in men [34]. Laboratory tests used or under discussion in patients with hemochromatosis are listed below (Table 3) [34,53,63,64].

Despite many reports on hemochromatosis disease, there is no quantitative diagnostic algorithm available, which could provide information on the probability grades of whether the diagnosis of hemochromatosis is more likely, possible, or excluded [33]. Various diagnostic approaches were published as algorithms, but none of these was based on evidence or quantitative results derived from scored key features [33,36,58,63,65,66,67,68,69,70,71] as opposed to other complex diseases such as autoimmune hepatitis (AIH) [72] or drug-induced liver injury (DILI) [73,74,75,76,77,78] with the Roussel Uclaf causality assessment method (RUCAM) published following international consensus meetings [75]. For AIH [72] and DILI [75], the scored diagnostic algorithms a used in line with artificial intelligence (AI) principles to ascertain the diagnosis.

AI represents a fascinating, provocative, and challenging discipline of pervasive and of global importance. The European Commission summarized the current state in a white paper on AI issues released on 19 February 2020, discussing various AI concepts that have revolutionized many complex processes [79]. The initial tools were algorithms, and, more recently, software programs are also used with an increasing tendency [79,80,81]. AI as a special term was created in 1956, when John McCarthy, a professor of mathematics at Dartmouth College, proposed a research project with the objective to simplify complex processes [80]. The principle was to provide tools enabling the input of data into a black box that systematically evaluates incoming data and fosters outputs of clear results such as diagnoses in complex diseases [81]. Only correct data should be given into the black box of AI that insures good data in and good data out. At the time when AI concepts were developed, the focus was on algorithms applied mostly manually prior to helpful software availability.

To establish a quantitative scored diagnostic algorithm of hemochromatosis, principles of artificial intelligence should be applied, which are based on the understanding that complex processes such as diagnoses in complicated diseases are best solved by introducing an algorithm based on individual items that correctly mirror disease characteristics. Using this approach, each element must be quantified by an individual score, and summing up the individual scores provides a final score with a degree of probability for the hemochromatosis diagnosis.

In general, a liver biopsy for histology examination is not required and not recommended anymore [63]. Instead, the use of magnetic resonance imaging (MRI) allows for iron detection and quantification not only in the liver but also in other organs. This approach has emerged as the reference standard imaging modality. Ultrasound is unable to detect iron overload, and computed tomography findings are nonspecific and influenced by multiple confounding variables [52].

### 5.6. Therapy

RCTs in patients with hemochromatosis are lacking because venesection syn phlebotomy was introduced early as a first line of therapy. Based on early theoretical considerations, this therapy was considered as the best way to efficiently remove excess iron from the body [9,10,63]. Further research is viewed as unlikely to change the confidence in the estimate of the benefit and risk of venesection in this disease [63]. Consensus among experts exists that venesection is the first-line therapy for patients with hemochromatosis with similar proposals for how best to proceed [63,82]. As an example of qualification for venesection, patients should have a serum ferritin level >300 μg/L in men and >200 μg/L in women in combination with an elevated fasting transferrin saturation (>45%) for both men and women. Weekly or fortnightly venesection aims to achieve an initial serum ferritin target of 50–100 μg/L [82]. Each venesection typically removes 500 mL of blood, corresponding to 250 mg of iron [82]. Venesection intervals are increased, or a lower volume of blood is removed if the patient does not tolerate phlebotomy because of weakness, hypotension, or the development of anemia [82]. Venesection ameliorates overall prognosis but must be conducted throughout the whole patient life but mostly with reduced intervals [9,10,63].

A recent randomized crossover trial focused on erythrocytapheresis. This therapeutic approach removes only red blood cells as opposed to phlebotomy, which removes blood with all cells [83]. The conclusion was reached that erythrocytapheresis reduces the number of annual treatment procedures but at higher costs compared with venesection in the maintenance treatment of patients with HFE hemochromatosis [83]. Subsequently, classified as personalized erythrocytapheresis, it is seen as the preferred treatment in selected cases [63].

As a second-line therapy in hemochromatosis for patients experiencing problems with venesection, iron chelation therapy using the once-daily oral iron chelator deferasirox (Exjade) is available in selected patients [84,85,86]. This chelator therapy is confined to patients with contraindications for venesection such as anemia, severe heart disease, or poor venous access [85]. However, this drug regime is associated with possible moderate digestive and renal side effects and is an off-label drug [86]. Several innovative pharmacotherapies are in progress under the evaluation of clinical trials. New drugs target key players that trigger iron overload in hemochromatosis with a focus on the specific causative intestinal iron absorption mechanism, hepcidin upregulation, and nanoparticles for iron chelation [13,86]. These novel approaches are opposed to symptomatic phlebotomy that only removes existing iron overload [86]. Among the more rational and causative therapeutic approaches were studies on hepcidin supplementation with a subcutaneous injection of synthetic hepcidin [86]. It was initially tested successfully in animals and healthy volunteers but was discontinued due to mixed results in a phase II study with hemochromatosis patients. On the other hand, ferroportin antagonists have not been studied in patients with hemochromatosis [86].

Liver transplantation (LT) was performed early in selected patients with hemochromatosis, but conflicting data on outcomes were reported [87,88,89,90]. However, it resulted in lower survival rates compared with patients transplanted for other terminal stages of liver diseases [88,89,90]. The reduced rates were mainly attributed to cardiac complications, fatal bacterial and fungal infections, and sepsis [88,89] or extrahepatic causes [90]. New data were used from the United Network for Organ Sharing registry with 862 adult patients listed for hemochromatosis from 2003 to 2019, but only 479 patients (55.6%) underwent LT [87]. Thereof, the 1-year post-LT survival rate in patients with hemochromatosis was 88.7% (95% confidence interval [CI], 85.4–91.4%), and the corresponding 5-year rate was 77.5% (95% CI, 72.8–81.4%). Both rates were comparable with those in the propensity-matched chronic liver disease (CLD) cohort [87]. This shows that short- and long-term survival rates for hemochromatosis are excellent and comparable with those of other LT recipients [87]. However, comparative data of the natural course of the 383 patients (44.4%) with hemochromatosis on the wait list, who did not undergo LT, were not provided. In addition, predictors for long-term (5-year) post-LT mortality included the presence of portal vein thrombosis with a hazard ratio (HR) of 1.96 and obesity measurements greater than class II with an HR of 1.98 at the time of LT. The leading cause of post-LT death (n = 145) was malignancy (25.5%), whereas cardiac disease was the cause in less than 10% of the recipients. Improving extrahepatic metabolic factors and functional status in patients with hemochromatosis prior to LT may improve outcomes [87].

As expected, LT cures patients with hemochromatosis because the new transplanted liver synthesizes hepcidin like in healthy individuals and reverts the genetic increased intestinal iron uptake [90,91]. Evidence for this curative effect was provided by only minimal iron deposits in liver biopsies obtained after successful LT [90].

### 5.7. Prognosis

Based on a large early cohort consisting of 163 patients with hemochromatosis before the era of liver transplantation, the cumulative survival rate was 92% at 5 years after diagnosis, 76% at 10 years, 59% at 15 years, and 49% at 20 years [9,10]. More specifically, life expectancy was lower in patients with cirrhosis as compared with those patients devoid of cirrhosis. Life expectancy was lower in patients with diabetes mellitus as compared with those without diabetes. It was also reduced in patients who could not be depleted of iron during the first 18 months of venesection therapy as compared with those who could be depleted [9,10]. Prognosis was not influenced by sex. In addition, patients without cirrhosis had a life expectancy that was similar to that expected in an age- and sex-matched normal population [9,10]. Compared with the normal population, the causes of death analyzed in 53 patients showed that hepatocellular carcinoma was 219 times more frequent among the patients (16 patients), cardiomyopathy was 306 times more frequent (3 patients), cirrhosis was 13 times more frequent (10 patients), and diabetes mellitus was 7 times more frequent (3 patients) [9,10]. Finally, death rates for other causes, including extrahepatic carcinomas (seven patients), were similar to the rates expected. The conclusion was reached that patients with hemochromatosis diagnosed in the pre-cirrhotic stage and treated by early venesection have a normal life expectancy, whereas cirrhotic patients have a shortened life expectancy and a considerable risk of liver cancer even when complete iron depletion has been achieved [9,10].

## 6. Mechanistic Steps Involved in the Iron Liver Injury

The pathogenetic steps of the cascade events involved in iron liver injury were analyzed and discussed in earlier review articles [5,6,34,92]. Although the focus primarily is on hemochromatosis, various aspects were derived from exogenous intoxications by overdosed iron in patients and animals. The cascade of events is listed in short to provide a quick overview (Table 4).

The first mechanistic step leading to liver injury found in patients with hemochromatosis (Table 4) is attributable to the genetic-based continued absorption of iron in excess far above demand for healthy life. Iron uptake occurs in humans through the intestinal tract [29] with a preference of the upper small intestine with its microvillous membrane vesicles [23]. This confirms results derived from a rat model [25,26]. The excessive intestinal uptake of iron leads to a toxicity of various organs, including the liver, in patients with hemochromatosis [9,10,11,12,31,63,95]. Hemochromatosis is an autosomal-recessive disorder caused by mutations in the genes involved in iron metabolism responsible for the abnormal increase in intestinal iron absorption [31] as a consequence of mutations in several genes, including HFE, transferrin receptor 2 (TFR2), hepcidin, ferroportin (SLC40A1), and hemojuvelin (HFE2) [31,33,34,92,96]. Most important is the mutation in the HFE gene, which encodes the HFE protein [31,34]. This protein normally regulates the production of hepcidin responsible for iron homeostasis and especially the control of iron uptake by cells through its interaction with transferrin receptors, processes that do not function in patients with hemochromatosis [34]. This is evidenced by the downregulation of hepcidin synthesis, leading to increased intestinal iron absorption [31]. As a result, mutations in the *HFE* gene lead to excess iron absorption and iron overload in hemochromatosis [52].

The second mechanistic step represents excessive iron uptake by hepatocytes from blood, where it is transported by transferrin [14,93]. This follows intestinal iron uptake as the first step of the cascade of events leading to liver injury [29].

The third step in hemochromatosis occurs in the liver itself with iron-overloaded hepatocytes. It starts with injurious attacks of iron as divalent ferrous iron (Fe^2+^), a cation capable of reacting with hydrogen peroxide generating one of the reactive oxygen species (ROS), the hydroxyl radical, while being oxidized to Fe^3+^ [14]. The radicals generated in the so-called Fenton–Haber–Weiss reaction are known as some of the most dominant oxidants found in the human body. Emerging radicals attack proteins, lipids, nucleic acids, and carbohydrates, leading to peroxidation and cell apoptosis [14,59,97,98,99]. There was the early notion that biologic iron in the presence of oxygen is toxic, but oxidative stress toxicity could not be entirely caused by the slow kinetics of the Haber–Weiss reaction. Consequently, stress toxicity must be finalized rather by the Fenton reaction, and Fritz Haber and his student Joshua Weiss advanced and postulated equations for iron-damaging reactions (Table 5) [59,99].

The Haber–Weiss reaction combined with the subsequent Fenton reaction leads to a net reaction with the generation of various ROS (Table 5), where ROS injure the membranes of subcellular organelles such as mitochondria or the endoplasmic reticulum [59]. They contain structural proteins and phospholipids with polyunsaturated fatty acids (PUFAs) that are peroxidized. This is evidenced by lipid peroxide markers found in patients with hemochromatosis or overloaded by exogenous iron [98,99,100,101,102]. As an example, in patients with hemochromatosis, liver biopsy specimens were immunostained for protein adducts with malondialdehyde and 4-hydroxynonenal [103]. Both adducts were found to be more abundant as compared with controls, whereby the staining had a predominance in acinar zone 1 that followed the localization of iron [103]. Similarly, enhanced oxidative stress was described in patients with hemochromatosis, as evidenced by hepatic malondialdehyde (MDA)-protein adducts and by increased oxidatively modified serum proteins [100]. MDA-lysine epitopes and oxidatively modified serum proteins, as well as immunoglobulin G autoantibodies against MDA-lysine epitopes, were increased in untreated hemochromatosis patients compared with normal individuals [100]. These markers of ongoing oxidative stress decreased with phlebotomy treatment in hemochromatosis patients [100]. After iron removal, there was a normalization of TGF-beta1 colocalized with hepatic iron and of MDA protein adducts in hepatocytes and sinusoidal cells of hepatic acinar zone 1. Iron overload increases both lipid peroxidation and TGF-beta1 expression, which together could promote hepatic injury and fibrogenesis, leading to liver fibrosis and finally cirrhosis [100].

Critical conditions emerge for patients with hemochromatosis who are confronted by a prolonged use of higher amounts of alcohol. This increases the hepatic iron content [60,61,62,103] and causes an inducted activity of the hepatic microsomal ethanol-oxidizing system (MEOS) [104,105] with CYP2E1 as its major constituent [106,107,108,109]. Prolonged alcohol abuse also increases reductase activity and the content of phospholipids as essential components of MEOS anchored in the membranes of the endoplasmic reticulum [105]. Polyunsaturated fatty acids (PUFAs) are structural components of membrane phospholipids and are easily peroxidized to form lipid peroxides, triggered by ROS. MEOS represents a pathway for metabolizing ethanol to acetaldehyde in a reaction requiring reduced nicotinamide adenine dinucleotide phosphate (NADPH) and molecular oxygen that is different from cytosolic alcohol dehydrogenase (ADH) [105]. Apart from producing the hepatotoxic acetaldehyde, in the presence of ethanol, MEOS also generates the ROS-like ethoxy radical CH_3_CH_2_O•, hydroxyethyl radical CH_3_C(•)HOH, acetyl radical CH_3_CHO•, single radical ^1^O_2_, superoxide radical HOO•, hydrogen peroxide H_2_O_2_, hydroxyl radical HO•, alkoxyl radical RO•, and peroxyl radical ROO• [108,109]. These byproducts can aggravate ROS-dependent liver injury in patients with hemochromatosis because antioxidant systems are reduced, as shown by decreased hepatic glutathione levels. Even worse, ethanol present nearby membranous CYP2E1 increases MEOS activity and thereby augments ROS generation with deleterious effects [110,111,112]. The reactions within the CYP cycle generate ROS as toxic intermediates and by-products, mostly after the uptake of two electrons and the incorporation of molecular oxygen that is then incompletely split [108]. ROS generated in the hepatic endoplasmic reticulum through the action of CYP 2E1 during microsomal ethanol metabolism create metabolic alterations and membrane injury of subcellular organelles through microsomal oxidative stress [108,109,110,111,112,113], which is to be differentiated from mitochondrial oxidative stress due to mitochondrial CYP 2E1 [114,115,116]. Since Fe^3+^ produces Fe^2+^ via the Haber–Weiss reaction, a vicious cycle develops because the subsequent Fenton reaction uses Fe^2+^ and thereby generates recycled Fe^3+^, which again can be used for the next cycle initiated by the Haber–Weiss reaction [116]. This combination of alcohol use and iron overload explains the deleterious effects on the liver injury. Similar to the Haber–Weiss reaction and the Fenton reaction providing an recycling of iron, substrates of cytochrome P450, like ethanol, also cause a recycling of Fe^3+^ via Fe^2+^ back to Fe^3+^, as evidenced within the catalytic cycle of CYP promoting the microsomal oxidation of ethanol as a substrate, as shown below schematically (Figure 2).

Ethanol metabolism via MEOS proceeds through the cytochrome P450 catalytic cycle (Figure 2) and the associated generation of ROS in the presence of various microsomal components (Figure 3).

Ferroptosis, the recently described iron-dependent form of regulated cell death [117,118,119,120], is closely related to mechanistic sequalae described in conditions of iron overload like hemochromatosis [119,120,121]. There is some evidence that ferroptosis is triggered by ferritinophagy, an autophagic process that specifically involves ferritin to release intracellular free iron [122]. At the morphological level, ferroptosis causes injury to mitochondria with a condensed ruptured outer membrane, associated with initial iron accumulation, excessive ROS production, and excessive lipid peroxidation [119]. Ferroptosis is also involved in alcoholic liver injury via liable iron accumulation and in the associated hepatic glutathione exhaustion [123].

The fourth step addresses the possible diagnostic or mechanistic role of cytokines in hemochromatosis. The liver is viewed as a secret-keeping organ, but it provides a few parameters like cytokines or chemokines in the blood for the quick analysis of mechanistic processes within the liver in the context of liver injury. In patients with hemochromatosis, several cytokines were analyzed, including IL1α, IL1β, IL2, IL4, IL6, IL8, IL10, IL12, IL17, IFNγ, TNFα, and Gm-CSF, which could help in investigating the inflammatory status of the study population. As opposed to none of the individuals in the control group, serum IL8 was elevated in 42% of C282Y homozygotes and 46% of H63D patients as homozygotes or in combination with C282Y [31]. The observation that several hemochromatosis patients had elevated levels of IL8 is difficult to explain but may be due to the fact that the C282Y HFE protein induces the transcription factor NF-κB, which consequently results in a marked increase in protein production of IL8 along with an increased transcriptional activation of IL8 in C282Y HFE-expressing cells [31,124]. Elevated levels of IL8 were especially observed in patients who had recently been treated with phlebotomy, which might suggest a relation between disease severity and levels of IL8 [31]. Similar to other cytokines, IL8, as a chemokine, is produced by macrophages and other cell types such as epithelial cells, airway smooth muscle cells [125], and endothelial cells, which retain IL-8 in their storage vesicles, known as Weibel–Palade bodies [126,127]. IL8 is also a major activating factor of neutrophils, and several neutrophil-secreted proteins were found to be significantly upregulated in patients harboring the H63D mutation [14,31]. Conceptually, however, the causes of neither the increased serum IL8 levels nor the unchanged levels of the other cytokines observed in the hemochromatosis patients were thoroughly investigated. As a consequence, these results currently do not add to our understanding regarding the mechanistic steps involving mediators that are secreted by non-parenchymal cells within the liver and may interact among each other [14,31]. The failure to detect many cytokines [31] may be ascribed to the fact that the respective studies were carried out in patients with only advanced stages of hemochromatosis already requiring phlebotomy therapy rather than in those with the early phases of the diseases. As an alternative, the immune cells that commonly secrete cytokines under normal conditions may have been so heavily injured by the iron not allowing anymore cytokine production and secretion.

Finally, the fifth mechanistic step is focused on the gut microbiome, which is altered in hemochromatosis patients [82,94]. Systemic iron reduction by phlebotomy was associated with an alteration in the gut microbiome, with changes evident in those who experienced reduced fecal iron availability with venesection [82]. For example, levels of *Faecalibacterium prausnitzii*, a bacterium associated with improved colonic health, were increased in response to fecal iron reduction. During iron depletion, iron absorption from the gastrointestinal tract increases to compensate for the iron lost with treatment. Consequently, iron availability is limited in the gastrointestinal tract and is crucial to the growth and function of many gut bacteria. Moreover, an increased colonic iron level has been associated with colonic inflammation and oxidative stress [82]. Similarly, metabolomic changes were seen in association with reduced fecal iron levels with significant changes in microbial metabolites after treatment, where increases in pyruvate, tyrosine, methionine, glycine, and aspartate were observed. In these patients, there was a greater separation in the metabolome [82]. A shift was observed towards a more positive metabolomic profile with treatment compared with the baseline [82]. There is also the note that iron availability for small-intestinal microbiota, specifically in the duodenum, likely differs from that for colonic microbiota, since small–intestinal microbiota are home to a lower density of residing microorganisms compared to the colon [94]. Excellent details on the aspects of ferroptosis have been provided in a recent report [128].

## 7. Alcohol

Alcohol abuse as a confounder and risk factor in hemochromatosis remains an important clinical and pathogenic issue, including in hepatic cell death and ferroptosis [129]. However, the specific involvement and regulatory mechanisms of ferroptosis in alcoholic liver disease remain poorly understood [123,129], also regarding miRNAs in regulating ferroptosis sensitivity. Novel insights into the involvement and regulatory mechanisms of ferroptosis in alcoholic liver disease as well as in hemochromatosis may highlighting the potential therapeutic implications of targeting ferroptosis and miRNAs in disease management [123].

## 8. Conclusions

Hemochromatosis is a genetic liver disorder based on overwhelming intestinal iron uptake because of mutations of the human hemochromatosis high-FE2+ (*HFE*) gene. For the development of this iron-induced liver disease, the following mechanistic steps within the cascade of events are proposed: (1) The first pathogenetic step in hemochromatosis causing liver injury can be traced back to genetically triggered overwhelming iron absorption from common iron-containing foods. (2) Excessive amounts of iron are then taken up by the hepatocytes from blood, where it is transported by transferrin. (3) The third step in hemochromatosis occurs in the liver itself in hepatocytes overloaded with iron, starting with injurious attacks of iron as divalent ferrous iron (Fe^2+^) through reaction with hydrogen peroxide and generating radicals, summarized as reactive oxygen species (ROS), which result during the Haber–Weiss reaction and the Fenton reaction. ROS attack proteins, lipids, nucleic acids, and carbohydrates, leading to lipid peroxidation and cell apoptosis. This may be associated with ferroptosis, an iron-dependent form of regulated cell death. (4) The fourth step addresses the possible role of cytokines like IL-8 in hemochromatosis. (5) Finally, the fifth mechanistic step has the focus on the gut microbiome, which is altered in hemochromatosis patients. In sum, hepatic iron overload is characteristic for hemochromatosis, whereby iron is toxic through generating ROS.

## Figures and Tables

**Figure 1 ijms-25-02668-f001:**
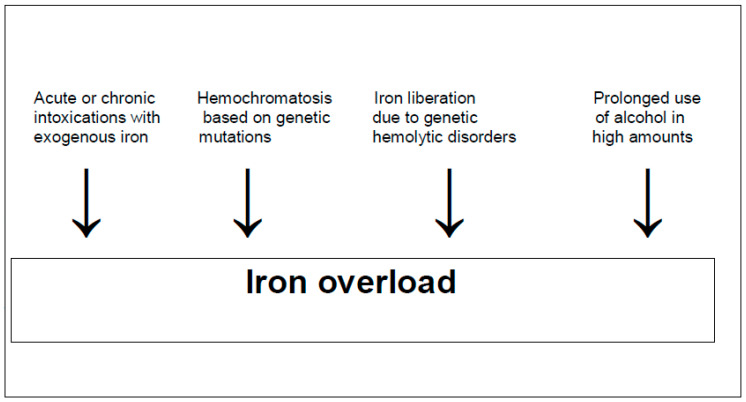
Causes of hepatic iron overload requiring careful clinical differentiation.

**Figure 2 ijms-25-02668-f002:**
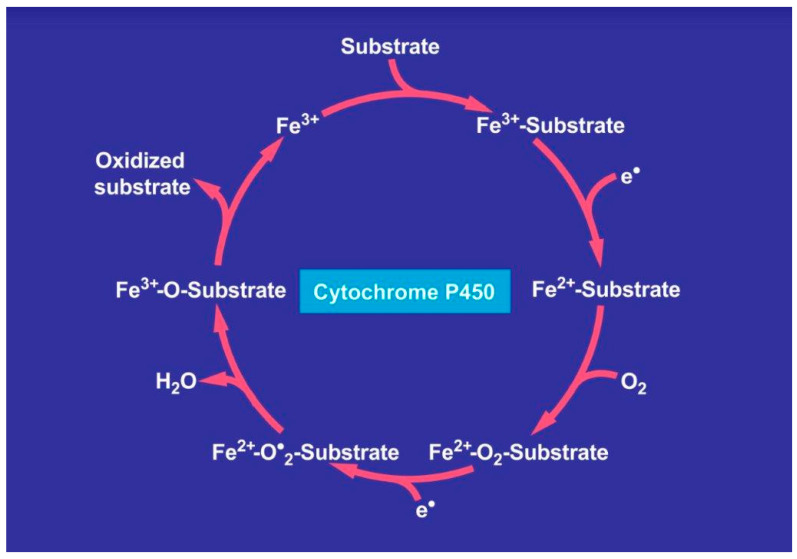
Catalytic cycle of cytochrome P450 and substrate interaction. Ethanol as a substrate of cytochrome P450 is metabolized following several mechanistic steps involving oxygen, electrons derived from NADPH + H+, and reactive oxygen species. The Fe^3+^ of the cytochrome is recycled via Fe^2+^ back to Fe^3+^. The original figure was published previously in an open-access article [108].

**Figure 3 ijms-25-02668-f003:**
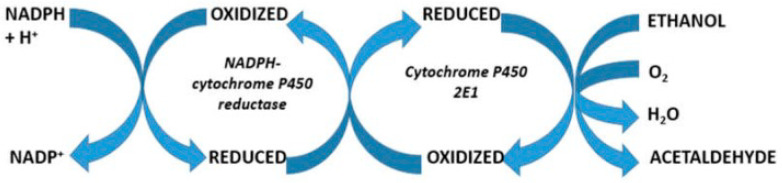
Hepatic microsomal metabolism of various substrates including ethanol. The reaction requires NADPH-cytochrome P450 reductase and cytochrome P450 with phospholipids as components facilitating the electron transfer [105,106,107,108,109,110]. This figure was derived from a previous open-access publication [106].

**Table 1 ijms-25-02668-t001:** List of iron’s contributions to human health, shown with selected examples.

Parameter	Contribution of Iron to Human Health	References
Hemoglobin	Around 80% of the total iron in body stores are found in the hemoglobin of erythrocytes. Iron is required for hemoglobin synthesis in the context of erythropoiesis, whereby erythroblasts in the bone marrow form erythrocytes responsible for oxygen transport. Iron is recycled from senescent erythrocytes and thus conserved by the body and stored by macrophages in spleen, liver, and bone marrow.	Roemhild et al., 2021 [13], Vogt et al., 2021 [14], Abbaspour et al., 2014 [15]
Myoglobin	Fe^2+^ is bound to a heme group of myoglobin, which helps bind oxygen reversibly. Myoglobin is a protein primarily found in the striated muscles and supplies the muscle oxygen to myocytes.	Abbaspour et al., 2014 [15]
DNA synthesis, nucleic acid repair	Iron is involved in DNA biosynthesis and is a known indispensable functional cofactor of helicases, nucleases, glycosylases, demethylases, and ribonucleotide reductase.	Roemhild et al., 2021 [13], Vogt et al., 2021 [14]
Cell growth	Iron is an essential element for the growth of all cells, whereby the rapid proliferation of tumor cells is usually more dependent on iron than normal cells are.	Roemhild et al., 2021 [13], Vogt et al., 2021 [14]
Host defense, cell signaling	Iron is essential for the host and pathogens in managing cellular and metabolic processes. Free iron, Fe^2+^, is involved in the Haber–Weiss reaction and the Fenton reaction that generate reactive oxygen species (ROS), supporting the host defense processes. Iron modulates immune cell function as well as the host-and-microbe interplay.	Vogt et al., 2021 [13]
Iron transporter proteins, heme enzymes, iron-containing enzymes	Iron is an essential part of iron transporter enzymes, heme enzymes, and other iron-containing enzymes involved in electron transfer and oxidation–reductions like cytochrome P450.	Vogt et al., 2021 [13], Abbaspour et al., 2014 [15]

Abbreviation: ROS, reactive oxygen species.

**Table 2 ijms-25-02668-t002:** Genetic details of hemochromatosis types.

Hemochromatosis Type	Details	First Author
Type 1 HFE-related	This is the classic form of hemochromatosis that is inherited in an autosomal-recessive fashion with a worldwide prevalence.	Bardou-Jacquet et al., 2014 [45], Yun et al., 2015 [46]
Type 2a Mutations in the hemojuvelin gene	Autosomal-recessive disorder that is seen both in whites and non-whites. Its onset is usually at 15–20 years.	Porter, 2023 [34]
Type 2b Mutations in the hepcidin gene	Autosomal-recessive disorder that is seen both in whites and non-whites. Its onset is usually at 15–20 years.	Porter, 2023 [34]
Type 3 Mutations in the transferrin receptor-2 gene	Autosomal-recessive disorder that is seen both in whites and non-whites. Its onset is at 30–40 years.	Joshi et al., 2015 [47]
Type 4 Mutations in the ferroportin gene	Autosomal-dominant disease seen both in whites and non-whites. Its onset is at 10–80 years.	Porter, 2023 [34]

**Table 3 ijms-25-02668-t003:** Laboratory test results in patients with hemochromatosis.

Laboratory Test	Normal Range	Test Details in Patients with Hemochromatosis	References with First Author
Serum ferritin	<15 µg/L	>1000 µg/L	Porter et al., 2023 [34], Daru et al., 2017 [53]
Fasting transferrin saturation index	<45%	>45%	EASL, 2022 [63]
Genetic screening	NA	*HFE* gene mutations	Porter et al., 2023 [34]
Serum ALT	<40 U/L	Usually < 80 U/L	Porter et al., 2023 [34]
Serum iron	Variable	Not suitable as a diagnostic marker	Grønlien et al., 2021 [31]
Serum juvelin	NA	Under discussion as a hepcidin regulator in hemochromatosis	Porter et al., 2023 [34], EASL, 2022 [63], Srole et al., 2021 [64]
Serum erythroferrone	NA	Under discussion as a suppressor of induced hepcidin in hemochromatosis	Srole et al., 2021 [64]

Abbreviations: NA, not available.

**Table 4 ijms-25-02668-t004:** Sequalae of events leading to the liver injury in patients with hemochromatosis.

Cascade of Events	Short Description	References with First Author
1. Excessive intestinal uptake of iron	Mutations in the *HFE* gene lead the downregulation of hepcidin synthesis to excess iron absorption and iron overload in hemochromatosis	Golfeyz et al., 2018 [52]
2. Uptake of high iron amounts by the liver cells from the blood	Intracellular iron initiates liver injury because the function of antioxidants is impaired due to low hepatic levels	Vogt et al., 2021 [14], Faruqi et al., 2023 [93]
3. Intracellular Fe^2+^ reacts with ROS and facilitates ferroptosis, an iron-dependent regulated cell death, causing liver injury through phospholipid peroxidation	ROS are generated via the Haber–Weiss and Fenton reactions and attack structural proteins, lipids, nucleic acids, and carbohydrates This leads, among others, to membranous phospholipid peroxidation. Liver injury is aggravated by alcohol abuse that increases hepatic iron levels and enhances ROS production via hepatic cytochrome P450 induction	Ali et al., 2023 [60], Li et al., 2022 [61], Louvet et al., 2015 [62]
4. Cytokines	Among the many mediators, including the interleukines (IL) tested, serum IL8 was elevated in patients with hemochromatosis. This indicated a close relationship of iron with hepatic macrophages, which retain IL8 in their storage vesicles	Grønlien et al., 2021 [31]
5. Gut microbiome	Systemic iron reduction by phlebotomy modifies the gut microbiome through the improvement in colonic inflammation and oxidative stress	Parmanand et al., 2020 [82], Yilmaz et al., 2018 [94]

Abbreviations: ROS, reactive oxygen species.

**Table 5 ijms-25-02668-t005:** Cascade of events triggered by the Haber–Weiss and Fenton reactions.

Details of the Haber–Weiss and Fenton Rection
Fe^3+^ + •O_2_^−^ → Fe^2+^ + O_2_ (Haber–Weiss reaction) Fe^2+^ + H_2_O_2_ → Fe^3+^ + OH^−^ + •OH (Fenton reaction) •O_2_^−^ + H_2_O_2_ → OH^−^ + •OH + O_2_ (Net reaction)

## Data Availability

Data are available in quoted publications.

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
