# Peer review of "Hemochromatosis: Ferroptosis, ROS, Gut Microbiome, and Clinical Challenges with Alcohol as Confounding Variable"

_ijms, 2024, doi:10.3390/ijms25052668_

Round 1

Reviewer 1 Report (Previous Reviewer 2)

Comments and Suggestions for Authors

The work has been revised and expanded significantly. The new information and more detailed bibliography make it a better review than it was before. Some smudges (deviating the reader from the main topic) have also been smoothed out. I believe the work can now be published.

Author Response

Done with attached file.

Reviewer 2 Report (Previous Reviewer 3)

Comments and Suggestions for Authors

The revised manuscript is massively improved in quality with a substantial portion being completely rewritten or new.  It is more focused; has a direction, and a small amount of novelty, which the original lacked.

My single and major concern is still the amount of minor editing required.  Here is a brief list:

1)  In the reference list, no text appears beside reference number 130.

2) A reference 131 appears in the reference list, but I cannot find it cited anywhere in the manuscript, which appears to end with refence 129.

3) I do not see why there is a need to mention it all, particularly multiple times, that plants and animals do not synthesize iron.  Of course, they cannot synthesize an element.  This very basic science need not be described multiple times.  Focus on the material of the review.

4) Table 1 - iron transporter enzymes should be iron transporter proteins at each use.

5) Line 98 - I believe intention should be intestines.

6) Carefully check the manuscript for typographical errors.  For example, line 300 has an extraneous "c".

7) Using your format, superoxide should be HOO., not HO.2.

8) Line 593 - what does "common iron containing food" mean (note you should have a hyphen between iron and containing)? Virtually all food has iron in significant amounts.  maybe just "food" here?

9) The kind of minor wording issues in 8) are all throughout the manuscript, which needs another round of very careful editing.

10) The manner in which alcohol abuse is now a focus (being mentioned in the title), I am surprised that there is not a specific section on alcohol as a confounding variable.  There are just scattered references to it throughout.  Can this be brought together better somehow?

Comments on the Quality of English Language

Greatly improved but many wording issues still exist.

Author Response

Done with attached file.

Reviewer 3 Report (Previous Reviewer 1)

Comments and Suggestions for Authors

The author tried to improve the manuscript following the reviewers' comments. Nevertheless, I still believe that the manuscript is not prepared with due care and does not contribute much to understanding the topic. Similar studies, but more carefully written, exist in the literature. Therefore, I still believe that the manuscript needs to be carefully re-edited.

Comments on the Quality of English Language

Minor editing of the English language is required.

Author Response

Done with attached file. 

Round 2

Reviewer 2 Report (Previous Reviewer 3)

Comments and Suggestions for Authors

Issues have been addressed.

Reviewer 3 Report (Previous Reviewer 1)

Comments and Suggestions for Authors

The Author have addressed my remarks in a thorough and satisfactory fashion. 

Comments on the Quality of English Language

Some minor changes.

This manuscript is a resubmission of an earlier submission. The following is a list of the peer review reports and author responses from that submission.

Round 1

Reviewer 1 Report

Comments and Suggestions for Authors

The manuscript covers an important topic: hemochromatosis, which is the most common human genetic disease. Despite the importance of the topic and the author's extensive clinical experience, the topic is treated carelessly and requires supplementation:

1. the author should clearly state whether the manuscript concerns primary or secondary hemochromatosi,

2. the description of the role of iron in the body is insufficient,

3. the author should explain the role of new biomarkers of iron metabolism in hemochromatosis, including hemojuvelin and erythroferrone, 

4. the manuscript suffers from a lack of figures and tables summarizing the topic,

5. the clinical part of the manuscript is well written.

Comments on the Quality of English Language

 Moderate editing of the English language is required.

Reviewer 2 Report

Comments and Suggestions for Authors

The work is presented as a review on hemochromatosis, a liver disease linked to the accumulation of iron.

However, the entire manuscript does not present itself as such. There is no mention of this intention in the title or abstract and, more importantly, the work is superficial, especially in the initial part. The information is not very relevant to the topic (repeated mention of the origin of atoms from stars!); or misleading (lines 44-46: it seems that cadmium is toxic because it exists in the environment and not because it is accumulated by human activities at toxic levels). 

section 2 state: Role of iron for human health. But when you go to read you talk about plants, cows, and very little about man. 

In addition, for being a refor a review paper there are very few references. for example, on lines 58-60 a list of functions is made but none are supported by a reference.

In Chapter 3 the author talks about deficiencies and overloads but there is not a single data on blood levels. 

the work must be enriched with more information, qualitative and quantitative data, completed to have a relevance for the reader

Comments on the Quality of English Language

sentences are sometimes too long but in all  English seems fine

Reviewer 3 Report

Comments and Suggestions for Authors

This manuscript is a review of the role of iron excess in hemochromatosis.  The manuscript is incredibly poorly written. The sentences are long, winding, and verbose, making the manuscript extraordinarily difficult to read.  Reading the manuscript would be most difficult for someone not a native speaker of English.  The entire manuscript needs to be rewritten with help from someone more familiar with writing scientific manuscripts in English.

Even then, reviews on this subject are numerous.  A search of PubMed for reviews on "iron and hemochromatosis" found dozens for just 2023.  The manuscript, thus, adds little if anything to the field.  Given the massive amount of reworking required to make it publishable and the lack of  new information provided, I cannot recommend publication.

Comments on the Quality of English Language

The manuscript is incredibly poorly written. The sentences are long, winding, and verbose, making the manuscript extraordinarily difficult to read.  Reading the manuscript would be most difficult for someone not a native speaker of English. 
